# Learning $k$-Level Structured Sparse Neural Networks Using Group Envelope Regularization

**Yehonathan Refael**                                            *refaelkalim@mail.tau.ac.il*
*Department of Electrical Engineering-Systems*
*Tel Aviv University*

**Iftach Arbel**                                            *i.arbel84@gmail.com*
*Independent Researcher*

**Wasim Huleihel**                                            *wasimh@tauex.tau.ac.il*
*Department of Electrical Engineering-Systems*
*Tel Aviv University*

**Reviewed on OpenReview:** *https://openreview.net/forum?id=XPLXYr7NlR*

## Abstract

The extensive need for computational resources poses a significant obstacle to deploying large-scale Deep Neural Networks (DNN) on devices with constrained resources. At the same time, studies have demonstrated that a significant number of these DNN parameters are redundant and extraneous. In this paper, we introduce a novel approach for learning structured sparse neural networks, aimed at bridging the DNN hardware deployment challenges. We develop a novel regularization technique, termed Weighted Group Sparse Envelope Function (WGSEF), generalizing the Sparse Envelop Function (SEF), to select (or nullify) neuron groups, thereby reducing redundancy and enhancing computational efficiency. The method speeds up inference time and aims to reduce memory demand and power consumption, thanks to its adaptability which lets any hardware specify group definitions, such as filters, channels, filter shapes, layer depths, a single parameter (unstructured), etc. The properties of the WGSEF enable the pre-definition of a desired sparsity level to be achieved at the training convergence. In the case of redundant parameters, this approach maintains negligible network accuracy degradation or can even lead to improvements in accuracy. Our method efficiently computes the WGSEF regularizer and its proximal operator, in a worst-case linear complexity relative to the number of group variables. Employing a proximal-gradient-based optimization technique, to train the model, it tackles the non-convex minimization problem incorporating the neural network loss and the WGSEF. Finally, we experiment and illustrate the efficiency of our proposed method in terms of the compression ratio, accuracy, and inference latency.

## 1 Introduction

In the past decade, significant progress has characterized the study of Deep Neural Networks (DNNs), which consistently demonstrate superior performance across the entire spectrum of machine learning tasks. As modern neural networks increase in size and complexity, with parameters often surpassing the number of available training samples, their deployment on resource-limited edge devices becomes increasingly challenging. This difficulty stems from the higher computational demands that lead to greater power consumption, longer inference times, and the need for substantial memory space for storage, which edge devices typically lack Deng et al. (2020); Cheng et al. (2017). Notwithstanding, many studies have revealed that modern neural networks tend to be excessively over-parametrized Han et al. (2015b); Ullrich et al. (2017). This over-parametrization implies the existence of redundant parameters that could be pruned (or, nullified)

without compromising network accuracy Molchanov et al. (2017), which are also responsible for issues such as overfitting Allen-Zhu et al. (2019), memorization of random patterns in the data Zhang et al. (2021), and a potential degradation in generalization. The realization that numerous redundant parameters exist has prompted a quest for neural network architectures that are both sparse and efficient, which emerges as a prominent challenge in the field.

To mitigate the challenges associated with the deployment of modern large DNNs, numerous studies have suggested compressing their scale. Various approaches have been explored, including (unstructured) sparsity including regularization Liu et al. (2015), pruning Han et al. (2015), low-rank approximation Han et al. (2015b); Denton et al. (2014a), quantization Gholami et al. (2022); Wu et al. (2021), and even sparse neural architecture search (NAS) Yang et al. (2020b); Wu et al. (2021). In the case of the unstructured sparsity-inducing, the most natural regularizer would be the so-called $\ell_0$-pseudo-norm function that counts the number of nonzero elements in the input vector, i.e., $\|\mathbf{z}\|_0 \triangleq |\{i : z_i \neq 0\}|$. These sparse regularized minimization/training problems are of the form $\min_{\mathbf{z} \in \mathbb{R}^n} \{f(\mathbf{z}) + \lambda\|\mathbf{z}\|_0\}$, or, alternatively, one can explicitly constrain the number of parameters used for regression and solve $\min_{\mathbf{z} \in \mathbb{R}^n} \{f(\mathbf{z}) : \|\mathbf{z}\|_0 \leq k\}$. Unfortunately, the $\ell_0$-norm is a difficult function to handle being non-convex and even non-continuous. Indeed, these types of regression models are known to be NP-hard problems, in general, Natarajan (1995) (global optimal solution can not be computed in a reasonable time, even for a very small number of parameters). As a remedy for the inherent problem above, Beck & Refael (2022) proposed a highly efficient tractable convex relaxation technique, termed sparse envelope function (SEF), for the sum of both $\ell_0$ and $\ell_2$ norms. Specifically, Beck & Refael (2022) suggested using this relaxation as a regularizer term for a convex loss objective, particularly for a linear regression model, to achieve feature selection while explicitly limiting the number of features to be a fixed parameter $k$. It was shown that the performance of this sparse inducing regularization method in both reconstruction of a sparse noisy signal and recovering its support, surpass the performance of state-of-the-art techniques, such as, the Elastic-net Zou & Hastie (2005), $k$-support norm Argyriou et al. (2012), etc. Also, it was shown that the computational complexity of the SEF approach is linear in the number of parameters, while all others require at least quadratic in the number of features, thus SEF was found very attractive.

Not long ago, the idea of structured sparsification was used in Wen et al. (2016); Bui et al. (2021) to learn sparse neural networks that leverage tensor arithmetic in dedicated neural processing units (NPUs). In a nutshell, structured sparsity learning amounts to inducing sparsity onto structured components (e.g., channels, filters, or layers) in the neural network during the optimization procedure. This leads, in practice, to both low latency and lower power consumption, which can not be obtained by deploying unstructured sparse models on such modern hardware.

With the goal of enabling structured sparsification learning that can be customized for different NPU devices, in this paper we propose a novel generalized notion of the SEF regularizer to handle group structured sparsification in neural network training. Our new generalized regularization term selects the most essential $k \leq m$ predefined groups of neurons (which could be convolutional filters, channels, individual neurons, or any other user-defined/NPU definition, where $m$ is the total number of groups) and prunes all others, while maintaining minimal network accuracy degradation. We define the new regularization term mathematically, propose an efficient method to calculate its value and proximal operator, and suggest a new algorithm to solve the complete optimization problem involving the non-convex term, which is the composition of the loss function and the neural network output.

**Related work.** The topic of regularization-based pruning received a lot of attention in recent years. Generally speaking, these studies can be divided into unstructured and structured pruning. Most prominent regularizers are the convex $\ell_1$ and $\ell_2$ norms Liu et al. (2017); Ye et al. (2018); Han et al. (2015b), as well as the non-convex $\ell_0$ "norm" Louizos et al. (2017b); Han et al. (2015); Louizos et al. (2017a), where Bayesian methods and additional regularization terms for practicality, were used to deal with the non-convexity of the $\ell_0$ norm. Additional works of Donoho & Elad (2003); Delmer et al. (2021) suggest methods for norm $l_0$ relaxation by employing $l_1$ minimization in general (nonorthogonal) dictionaries and leading to an error surface with fewer local minima than the $l_0$ norm. The motivation for these regularizers is their "sparsity-inducing" property which can be harnessed to learn sparse neural networks. While these fundamental papers significantly reduce the storage needed to store the networks on hardware, there were no benefits in reducing

the inference latency time or either in cutting down power consumption. That is, the sparse neural networks, learned by the aforementioned methods, were not adapted to the tensor arithmetic of the hardware they aimed to run on.

The practical inefficiency of unstructured sparsity-inducing methods has led researchers to propose regularization-based structured pruning in favor of accelerating the running time. For example, Lebedev & Lempitsky (2015); Wen et al. (2016); Yuan & Lin (2006) proposed the use of the Group Lasso regularization technique to learn sparse structures, and Scardapane et al. (2017) uses Sparse Group Lasso, summing Group Lasso with the standard Lasso penalty. Other convex regularizers include the Combined Group and Exclusive Sparsity (CGES) Yoon & Hwang (2017), which extends Exclusive Lasso (in essence, squared $\ell_1$ over groups) Zhou et al. (2010) using Group Lasso. Recently, Bui et al. (2021) suggested a family of nonconvex regularizers that blend Group Lasso with nonconvex terms ($\ell_0$, $\ell_1 - \ell_2$ Lou et al. (2015), and SCAD Fan & Li (2001)). Since Bui et al. (2021) introduces non-convexity term into the penalty, it also requires an appropriate optimization scheme, for which the authors propose an Augmented Lagrangian type method. However, this optimization algorithm has an inner optimization loop with a high computational cost. Moreover, their extensive experiments do not show an accuracy or sparsity advantages over convex penalties, suggesting that it might be still desirable to use a convex regularizer. Other methods, such as Chen et al. (2021); Li et al. (2019), focus on a group structure that captures the relations between parameters, neurons, and layers, in order to construct groups that can maximize network compression while minimizing accuracy loss. However, these methods still apply Group Lasso regularization. Specifically in Chen et al. (2021), the authors introduce the concept of Zero-Invariant Groups (ZIGs), which includes all input and output connections between layers. In the context of CNNs, it extends the channel-wise grouping Wen et al. (2016) to include corresponding batch normalization parameters. By using this group structure, entire blocks of parameters can be removed while keeping dimensions aligned between layers, and ultimately allowing network compression. Moreover, their optimization scheme utilizes a two-phase algorithm to include a half-space projection step, which they name HSPG. Lately, a novel methodology that applies adaptive optimization via weighted proximal operators to induce structured sparsity was suggested in Deleu & Bengio (2023) in seamlessly integrating numerical solvers to preserve convergence guarantees, albeit with computational efficiency concerns due to approximation requirements.

Finally, we mention that there exist other techniques for neural net compression, such as, quantization, low-rank decomposition, to name a few. In quantization, Courbariaux et al. (2016); Rastegari et al. (2016); Gong et al. (2014), the precision of the weights is reduced, by representing weights using a low number of bits (i.e., 8-bit) instead of higher one (i.e., 32-bit floating point values). The low-rank decomposition approach Denton et al. (2014b); Jaderberg et al. (2014); Lebedev et al. (2014); Li et al. (2020b) is based on the observation that many weight matrices in neural networks are highly correlated and can be well approximated by matrices with a lower rank. By decomposing a weight matrix into lower-rank matrices, one can reduce the total number of parameters in the network.

**Notation.** We denote $\mathbf{e}$ for the vector of all ones. For a positive integer $m$, we denote $[m] \equiv \{1, 2, \ldots, m\}$. We denote by $x_{\langle i \rangle}$ the component of $x$ with the $i$ the largest absolute value, meaning in particular that $|x_{\langle 1 \rangle}| \geq |x_{\langle 2 \rangle}| \geq \ldots \geq |x_{\langle n \rangle}|$. $|S|$ refers to the number of elements in the set $S$. Let $f : \mathbb{R}^n \to \mathbb{R}$, be an extended real-valued function, then, the conjugate function of $f$, denoted by $f^\star : \mathbb{R}^n \to \mathbb{R}$ is defined as $f^\star(y) = \max_{x \in \mathbb{R}^n} \{\langle x, y \rangle - f(x)\}$, for any $y \in \mathbb{R}^n$. The bi-conjugate function is defined as the conjugate of the conjugate function, i.e., $f^{\star\star}(x) = \max_{y \in \mathbb{R}^n} \{\langle x, y \rangle - f^\star(y)\}$, for any $x \in \mathbb{R}^n$. Finally, the proximal operator of a proper, lower semi-continuous convex function $f : \mathbb{R}^n \to \mathbb{R}$ is defined as $\text{prox}_f(v) = \arg\min_{x \in \mathbb{R}} \{f(x) + \frac{1}{2}\|x - v\|_2^2\}$, for any $v \in \mathbb{R}^n$. The sets $\mathbb{R}_+$ and $\mathbb{R}_{++}$ denote all non-negative and positive real numbers, respectively.

## 2 Structured sparsity via WGSEF

### 2.1 Problem formulation

In this subsection, we formulate the problem and introduce the weighted group sparse envelop function (WGSEF). Without loss of generality, our method is formulated on weights sparsity, but it can be directly

extended to neuron sparsity (i.e., both weights and bias). Let $\mathcal{D}$ be a dataset consisting of $N$ i.i.d. input output pairs $\{(x_1, y_1), \ldots, (x_N, y_N)\}$. The general neural network training problem is formalized as the following regularized empirical risk minimization procedure on the parameters $\theta \in \Theta$ of a given neural network architecture $f(\cdot; \theta)$,

$$\operatorname*{argmin}_{\theta \in \Theta} \frac{1}{N} \sum_{i=1}^{N} \mathcal{L}(f(x_i, \theta), y_i) + \lambda \cdot \Omega(\theta), \tag{1}$$

where, $f(\cdot; \theta)$ is the hypothesis, that is, a given neural network architecture, $\mathcal{L}(\cdot) \geq 0$ corresponds to a loss function, e.g., cross-entropy loss for classification, mean-squared error for regression, etc, $\Omega(\cdot) : \Theta \to \mathbb{R}_+$ is the parameters regularization term, $\lambda \in \mathbb{R}_+$ is the regularization magnitude. Below, $n \triangleq |\Theta|$ denotes the number of parameters.

The most predominant regularizer used for DNNs is the weight decay, also known as the $\ell_2$ norm regularization. It is known to prevent overfitting and to improve generalization since it enforces the weights to decrease proportionally to their magnitudes. The most natural way to force a predefined $k$-level sparsity would be to constrain the number of non-zeros parameters (e.g., the model weights), which can be done by adding the constraint that $\|\theta\|_0 \leq k$, where $k \leq n$ is the required predefined level of sparsity. In this case, the training problem is formalized as follows,

$$\operatorname*{argmin}_{\theta \in \Theta} \frac{1}{N} \sum_{i=1}^{N} \mathcal{L}(f(x_i, \theta), y_i) + \frac{\lambda}{2} \|\theta\|_2^2$$
$$\text{s.t.} \quad \|\theta\|_0 \leq k. \tag{2}$$

We refer to the above training problem as *unstructured sparsification*. In the case of *structured sparsity*, the parameters $\theta$ are divided into *predefined* disjoint sub-groups. These subgroups could define the building blocks architecture of DNNs, i.e., filters, channels, filter shapes, and layer depth. Consider the following definition.

**Definition 1** (Group projection). *Let $s$ be a subset of indexes $s \subseteq \{1, 2, \ldots, n\}$ of size $|s| \leq n$. Then, given some vector $\theta \in \mathbb{R}^n$, the projection $M_s : \mathbb{R}^n \to \mathbb{R}^n$ preserves only the entries of $\theta$ that belong to the set $s$. Furthermore, let $A_s$ be an $n \times n$ diagonal matrix, where $[A_s]_{ii} = 1$ if $i \in s$, and zero, otherwise. Note that $M_s(\theta) = A_s\theta$.*

**Example 1.** *Let $n = 3, \theta = (3, 6, 9)^\top, s = \{1, 3\} \subseteq [n]$, and accordingly $|s| = 2$, then $M_s(\theta) = M_s((3, 6, 9)^\top) = (3, 0, 9)^\top$, with $[A_s]_{11} = [A_s]_{33} = 1$, and zero otherwise.*

Following the above definition, let $m \leq n$ subsets $s_1, s_2, \ldots, s_m$ be a given (non-overlapping) partition of $[n]$, namely, $s_i \cap s_j = \emptyset$, for all $i \neq j$, and $\bigcup_{i=1}^{m} s_i = [n]$. Without loss of generality, we assume that $n \bmod m = 0$; otherwise, the groups would have different coordinates. Every group is associated with some weight $d_j \in \mathbf{R}_{++}$, where $j \in [m]$, e.g., $d_j = \frac{1}{|s_j|}$, namely, we normalize by the group size. For simplicity of notation, let $\theta s_i = M_{s_i}(\theta)$, for $i = 1, 2, \ldots, m$. Then, our structured training problem is,

$$\operatorname*{argmin}_{\theta \in \Theta} \frac{1}{N} \sum_{i=1}^{N} \mathcal{L}(f(x_i, \theta), y_i) + \frac{\lambda}{2} \sum_{j=1}^{m} d_j \|\theta s_j\|_2^2$$
$$\text{s.t.} \quad \left\| \|\theta s_1\|_2^2, \|\theta s_2\|_2^2, \ldots, \|\theta s_m\|_2^2 \right\|_0 \leq k. \tag{3}$$

To wit, we constrain the number of groups which has at least one non-zero coordinate, to be at most $k$. Let $C_k$ denote the set of all $k$ sparse groups, i.e.,

$$C_k \triangleq \left\{ \theta : \left\| \|\theta s_1\|^2, \|\theta s_2\|^2 \ldots, \|\theta s_m\|^2 \right\|_0 \leq k \right\},$$

and define $\delta_{C_k}$ as the following extended real-valued function,

$$\delta_{C_k}(\theta) \triangleq \begin{cases} 0, & \left\| \|\theta s_1\|^2, \|\theta s_2\|^2 \ldots, \|\theta s_m\|^2 \right\|_0 \leq k, \\ \infty, & \text{else.} \end{cases}$$

Then, the optimization problem in equation 3 can be reformulated as,

$$\underset{\theta \in \Theta}{\text{argmin}} \; \frac{1}{N} \sum_{i=1}^{N} \mathcal{L}\left(f\left(x_i, \theta\right), y_i\right) + \lambda \cdot gs_k(\theta), \tag{4}$$

where $gs_k(\theta) \triangleq \frac{1}{2} \sum_{j=1}^{m} d_j \|\theta s_j\|_2^2 + \delta_{C_k}(\Theta)$. Equivalently, $gs_k(\theta)$ can be rewritten as $gs_k(\theta) \triangleq \frac{1}{2} \sum_{i=1}^{n} d_i \cdot \theta_i^2 + \delta_{C_k}(\Theta)$, where $d_i = d_j$ for every $i \in s_j$.

The $\ell_0$-norm that appears in 4, is a difficult function to handle being nonconvex and even non-continuous, making the problem an untraceable combinatorial NP hard problem Natarajan (1995). Following the work in Beck & Refael (2022), one approach to deal with this inherent difficulty is to consider the best convex underestimator of $gs_k(\cdot)$. The later is its bi-conjugate function, namely, $\mathcal{GS}_k(\theta) = gs_k^{\star\star}(\theta)$, which we refer to as the *weighted group sparse envelope function (WGSEF)*.

**Remark 1** (Generalization of SEF). *Consider the case $m = n$, namely, every subset $s_i, i \in [m]$ is a singleton and $\forall j \in [m], d_j = 1$. Here, $gs_k(\theta) = s_k(\theta)$, where $s_k(\theta) = \frac{1}{2}\|\theta\|_2^2$ if $\|\theta\|_0 \leq k$ and $s_k(\theta) = \infty$, otherwise. Accordingly, in this case, $\mathcal{GS}_k(\theta) = \mathcal{S}_k(\theta) = s_k^{\star\star}(\theta)$, and $s_k^{\star\star}(\theta)$ is exactly the classical SEF, namely, $\mathcal{S}_k(\cdot)$. Therefore, $\mathcal{GS}_k(\cdot)$ is indeed a new generalization of SEF to handle group sparsity.*

Thus, the path taken in this paper is to consider the following relaxed learning problem (training),

$$\underset{\theta \in \Theta}{\text{argmin}} \; \frac{1}{N} \sum_{i=1}^{N} \mathcal{L}\left(f\left(x_i, \theta\right), y_i\right) + \mathcal{GS}_k(\theta). \tag{5}$$

In the following subsection, we develop an efficient algorithm to calculate the value and the prox-operator Beck (2017) of WGSEF; these will play an essential ingredient when solving (5).

## 2.2 Convex relaxation

Let us start by introducing some notation. For any $\theta \in \mathbb{R}^n$ and $m$ subgroups of indexes $s_1, s_2, \ldots, s_m \subset [n]$, we denote by $M_{\langle s_i \rangle}(\theta)$ the corresponding subgroup of coordinates in $\theta$ with the $i$th largest $\ell_2$-norm, i.e. $\forall i \neq j \in [m]$,

$$\left\|M_{\langle s_1 \rangle}(\theta)\right\|_2 \geq \left\|M_{\langle s_2 \rangle}(\theta)\right\|_2 \geq \ldots \geq \left\|M_{\langle s_m \rangle}(\theta)\right\|_2.$$

We next show that the conjugate of the $k$ group sparse envelopes is the $k$ weighted group hard thresholding function. In the sequel, we let $D$ be the $n \times n$ diagonal positive-definite weights matrix, such that $D_{i,i} = \sqrt{d_i}, \forall i \in s_j$, and $j \in [m]$.

**Lemma 1** (The $k$ weighted group sparse envelop conjugate). *Let subsets $s_1, s_2, \ldots, s_m$ be a set of $m \leq n$ disjoint indexes that partition $[n]$. Then, for any $\tilde{\theta} \in \mathbb{R}^n$,*

$$gs_k^{\star}(\tilde{\theta}) = \frac{1}{2} \sum_{j=1}^{k} \frac{1}{d_j} \left\|M_{\langle s_j \rangle}(\tilde{\theta})\right\|_2^2. \tag{6}$$

Next, we obtain the bi-conjugate function of the $k$ weighted group sparse envelope. To express the following results explicitly, we will deliberately utilize the group projection definition 1 expressed as $M_s(\theta) = A_s\theta$, for some set of indices $s$.

**Lemma 2** (The variational bi-conjugate $k$ weighted group sparse envelop). *Let $s_1, s_2, \ldots, s_m$ be a set of $m \leq n$ disjoint subsets that partition $[n]$. Then, for any $\theta \in \mathbb{R}^n$, the bi-conjugate of the $k$ group sparse envelop is given by*

$$\mathcal{GS}_k(\theta) = \frac{1}{2} \min_{\mathbf{u} \in B_k} \left\{ \sum_{j=1}^{m} d_j \phi\left(A_{s_j}\theta, u_j\right) \right\}, \tag{7}$$

*where,*

$$\phi\left(A_{s_j}\theta, u_j\right) \triangleq \begin{cases} \frac{\theta^\top A_{s_j}\theta}{u_j}, & u_j > 0, \\ 0, & u_j = 0 \cap A_{s_j}\theta = 0, \\ \infty & \text{else.} \end{cases} \tag{8}$$

The following is a straightforward corollary of Lemma 2.

**Corollary 2.1.** *The following holds:*

$$\mathcal{GS}_k(\theta) =$$
$$\mathcal{S}((\sqrt{d_1}\|A_{s_1}\theta\|_2, \sqrt{d_2}\|A_{s_2}\theta\|_2, \ldots, \sqrt{d_m}\|A_{s_m}\theta\|_2)^\top),$$

*where $\mathcal{S}(\theta) \triangleq s_k^{\star\star}(\theta)$ is the standard SEF.*

The above corollary implies that in order to calculate $\mathcal{GS}_k(\theta)$ we only need to apply an algorithm that calculates the SEF at $(\sqrt{d_1}\|A_{s_1}\theta\|_2, \sqrt{d_2}\|A_{s_2}\theta\|_2, \ldots, \sqrt{d_m}\|A_{s_m}\theta\|_2)^\top$.

**Remark 2.** *Noting $\|\sqrt{d_j}A_{s_j}\theta\|_2^2 = \sum_{i\in s_j} d_j\theta_i^2$ we observe that since $s_j$ is given, the number of operations required to calculate $\|\sqrt{d_j}A_{s_j}\theta\|_2^2$ is linear w.r.t. $|s_j|$. Thus, the computational complexity of calculating the vector $(\sqrt{d_1}\|A_{s_1}\theta\|_2, \sqrt{d_2}\|A_{s_2}\theta\|_2, \ldots, \sqrt{d_m}\|A_{s_m}\theta\|_2)^\top$ is linear in $n$.*

## 2.3 Proximal mapping of the WGSEF

In this subsection, we will show how to efficiently compute the proximal operator of positive scalar multiples of $\mathcal{GS}_k$. The ability to perform such an operation implies that it is possible to employ fast proximal gradient methods to solve equation 5. We begin with the following lemma that shows that the proximal operator can be determined in terms of the optimal solution of a convex problem that resembles the optimization problem defined in Lemma equation 7 for computing $\mathcal{GS}_k$.

**Lemma 3.** *Let $\lambda > 0$, $t \in \mathbb{R}^n$, and $s_1, s_2, \ldots, s_m$ be a set of $m \le n$ disjoint subsets that partition $[n]$. Then, $v = \text{prox}_{\lambda\mathcal{GS}_k}(t)$ is given by*

$$j \in [m], \quad A_{s_j}v = \frac{u_j A_{s_j}t}{\lambda d_j + u_j},$$

*where $(u_1, u_2, \ldots, u_n)^T$ is the minimizer of*

$$\min_{\mathbf{u}\in D_k} \sum_{j=1}^m \phi\left(\sqrt{d_j}A_{s_j}t, \lambda d_j + u_j\right). \tag{9}$$

Next, we show that the proximal operator of $\mathcal{GS}_k$ reduces to an efficient one-dimensional search.

**Corollary 3.1** (The proximal operator of $\mathcal{GS}_k$). *The solution $u_j = u_j(\mu^*)$ of equation 9 with $u_j(\cdot)$ defined as[1]*

$$u(\mu^*) = \begin{cases} 1, & \sqrt{\mu^*} \le \frac{|b_j|}{\alpha_j+1}, \\ \frac{|b_j|}{\sqrt{\mu^*}} - \alpha_j, & \frac{|b_j|}{\alpha_j+1} < \sqrt{\mu^*} < \frac{|b_j|}{\alpha_j}, \\ 0, & \sqrt{\mu^*} \ge \frac{|b_j|}{\alpha_j}. \end{cases} \tag{10}$$

*for $b_j = \left\|\sqrt{d_j}A_{s_j}t\right\|_2$ and $\alpha_j = \lambda d_j(> 0)$, and $\tilde{\eta} = \frac{1}{\sqrt{\mu^*}}$ is a root of the function*

$$g_t(\eta) \equiv \sum_{i=j}^m u_j(\eta) - k, \tag{11}$$

---

[1]If $A_{s_j}t = 0$, then equation 10 implies that $u_i(\mu) = 0$ for all $\mu \ge 0$.

*which is nondecreasing and satisfies*

$$g_t \left( \frac{\lambda \cdot \min_{j \in [m]}\{d_j\}}{\left\| \sqrt{d_1} \, \|A_{s_1} t\|_2 \,, \sqrt{d_2} \, \|A_{s_2} t\|_2 \,, \ldots, \sqrt{d_m} \, \|A_{s_m} t\|_2 \right\|_\infty} \right)$$
$$= \sum_{i=1}^m 0 - k < 0,$$

*and,*

$$g_t \left( \frac{\lambda \, \|d_1, d_2, \ldots, d_m\|_\infty + 1}{\left\| M_{\langle s_m \rangle} \left( \sqrt{d_j} t \right) \right\|_2} \right) = \sum_{i=1}^m 1 - k > 0.$$

*In addition, $g_t$ can be reformulated as the sum of pairs of the functions*

$$v_i(\eta) \equiv |\eta|b_j| - \alpha_j|, w_i(\eta) \equiv 1 - |\eta|b_j| - (\alpha_j + 1)|, j \in [m],$$

*such that,*

$$g_{\mathbf{t}}(\eta) = \frac{1}{2} \sum_{j=1}^m v_j(\eta) + \frac{1}{2} \sum_{j=1}^m w_j(\eta) - k.$$

The following important remarks are in order.

**Remark 3** (Root search application for function 11). *Employing the randomized root search method in (Beck & Refael, 2022, Algorithm 1) with the $2m$ one break point piece-wise linear functions $v_j, w_j$, as an input to the algorithm, the root of $g_{\mathbf{t}}$ can be found in $O(m)$ time.*

**Remark 4** (Computational complexity of $\mathrm{prox}_{\lambda \mathcal{GS}_k}$). *The computation of $\mathrm{prox}_{\lambda \mathcal{GS}_k}$ boils down to a root search problem (see, Remark 3), which requires $O(m)$ operations. In addition, before employing the root search, the assembly of $v_j, w_j$, requires the calculations of the $m$ values of $b_j$ defined in Corollary 11. Note that for any $j \in [m]$ calculating $b_j$ is equivalent to $t^\top A_{s_j} t = \sum_{i \in s_j} t_i^2$. Since $s_j$'s are given, the computational complexity of calculating all $m$ of $b_j$ is linear in $n$, which is the dimension of $t$. Thus, the total computational operations of calculating $\mathrm{prox}_{\lambda \mathcal{GS}_k}$ summarizes to $n$ with is the dimension of all groups parameters together.*

## 3 Optimization procedure

The general training problem we are solving is of the form

$$\min_{\mathbf{x} \in \mathbb{R}^n} F(x) = f(\mathbf{x}) + h(\mathbf{x}), \tag{12}$$

where $f = \frac{1}{N} \sum_{i=1}^N f_i : X \to \mathbb{R}$ is continuously differentiable, but possibly nonconvex, and $h$ is a convex function, but nonsmooth. We adopt the ProxGen Yun et al. (2021), which can accommodate momentum, and also has a proven convergence rate with a fixed and reasonable minibatch size of order $\Theta(\sqrt{N})$ (a comprehensive discussion on the selection of the optimization method is provided in the appendix A.4). Next, we provide a convergence guarantee for Algorithm 1, as given in Yang et al. (2020a). This result holds under several regularity assumptions which can be found in Appendix A.4.1.

**Corollary 3.2.** *Under Assumptions* (C-1)–(C-3), *Algorithm 1 with constant minibatch size $b_t = b = \Theta(T)$ is guaranteed to yield $\mathbb{E}\left[\mathrm{dist}\left(\mathbf{0}, \widehat{\partial} F(\theta)\right)^2\right] \leq O\left(T^{-1}\right)$, where $\widehat{\partial} F$ is the Fréchet sub-differential function of $F$.*

Forthwith, we propose Algorithm 2, as an implementation of Algorithm 1 to solve equation 5, where $f \triangleq \mathcal{L}$ and $h \triangleq \mathcal{GS}_k$. Calculating $\nabla \mathcal{L}(\boldsymbol{\theta}_t; \xi_t)$, commonly approached using a propagation algorithm, is at least linear in the number of parameters and is obviously getting more complex as the number of layers increases. Therefore, the calculation of the prox of the WGSEF is not a bottleneck of the update step complexity (since

---

**Algorithm 1: General Stochastic Proximal Gradient Method**

---

**Input:** Stepsize $\alpha_t, \{\rho_t\}_{t=1}^{t=T} \in [0, 1)$, regularization parameter $\lambda$.
**Initialization:** $\boldsymbol{\theta}_1 \in \mathbb{R}^n$ and $\mathbf{m}_0 = \mathbf{0} \in \mathbb{R}^n$.

**for** iteration $t = 1, \ldots, T$:
     Draw a minibatch sample $\xi_t$
     $\mathbf{g}_t \longleftarrow \nabla f(\boldsymbol{\theta}_t; \xi_t)$
     $\mathbf{m}_t \longleftarrow \rho_t \mathbf{m}_{t-1} + (1 - \rho_t)\mathbf{g}_t$
     $\boldsymbol{\theta}_{t+1} \longleftarrow \mathrm{prox}_{\alpha_t \lambda h} (\boldsymbol{\theta}_t - \alpha_t \mathbf{m}_t)$
**return** $\boldsymbol{\theta}$

---

---

**Algorithm 2: Learning structured $k$-level sparse neural-network by Prox SGD with WGSEF regularization**

---

**Input:** Stepsize $\alpha_t, \{\rho_t\}_{t=1}^{t=T} \in [0, 1)$, regularization parameters $\{\lambda_l\}_{l=1}^{l=L} \in [0, \infty)$.
**Initialization:** Randomly initialize the weights $\theta_{t=0} \in \mathbb{R}^n$, $\mathbf{m}_0 = \mathbf{0} \in \mathbb{R}^n$.

**for** iteration $t = 1, \ldots, T$:
     Draw a minibatch sample $\xi_t$
     $\mathbf{g}_t \longleftarrow \nabla \mathcal{L}(\boldsymbol{\theta}_t; \xi_t)$
     $\mathbf{m}_t \longleftarrow \rho_t \mathbf{m}_{t-1} + (1 - \rho_t)\mathbf{g}_t$
     $\boldsymbol{\theta}_{t+1} \longleftarrow \mathrm{prox}_{\alpha_t \lambda \mathcal{GS}_k} (\boldsymbol{\theta}_t - \alpha_t \mathbf{m}_t)$
**Prune (Optional):** all $\#(m - k)$ smallest $\ell_2$ group norm values of $\boldsymbol{\theta}$.
**return** $\boldsymbol{\theta}$

---

it is linear in the number of parameters). Notice that in our setting, one option is that the regularization may be separable by layer, as indicated by the regularization function's definition $h(\mathbf{x}) = \sum_{l=1}^{L} h_l(\mathbf{x}_l)$. Therefore, in this case, prox is applied to each layer separately (Beck, 2017, Theorem 6.6) with different $\lambda_l, k_l$ parameters per layer. Another option is that the regularization is applied to all groups' layers collectively, namely, not in a per-layer fashion, to allow more flexibility in group selection. Formally, in this latter option, only a single pair of $\lambda, k$ values needs to be selected, so $\forall l \in [L], \lambda_l = \lambda, k_l = k$. For example, regularization is applied simultaneously on filters across all convolutional layers at once, resulting in different filter sparsity levels in each layer, but overall adhering to the pre-defined filter sparsity level $k$. The only condition for both options is that groups are not overlapping. We note that since Assumptions (C-1)–(C-3) are met, Algorithm 2 converges to an $\epsilon$-stationary point.

Another technique for solving equation 12 is using the HSPG family of algorithms in Chen et al. (2020); Dai et al. (2023). These algorithms utilize a two-step procedure in which optimization is carried by standard first-order methods (i.e., subgradient or proximal) to find an approximation that is "sufficiently close" to a solution. This step is followed by a half-space step that freezes the sparse groups and applies a tentative gradient step on the dense groups. Over these dense groups, parameters are zeroed-out if a sufficient decrease condition is met, otherwise, a standard gradient step is executed. Notice that the half-space step, as a variant of a gradient method, requires the regularizer term $h$ to have a Lipschitz continuous gradient, which is not satisfied in our setting. However, this property is required only for the dense groups as there is no use of the gradient in groups that are already sparse. Since the continuity is violated only for sparse groups, the condition is satisfied in the required region. Finally, while the "sufficiently close" condition mentioned above cannot be verified in practice, simple heuristics to switch between steps still work well. We can either run the first-order step for a fixed number of iterations before switching to the half-space step, or, alternatively, run the first-order step until the sparsity level stabilizes, and then switch to the half-space step. The dense groups are defined as $\mathcal{I}^0(\mathbf{x}) := \{\gamma \,|\, \gamma \in \mathcal{G}, \|\mathbf{x}^\gamma\| = 0\}$, and sparse groups are defined as $\mathcal{I}^{\neq 0} := \{\gamma \,|\, \gamma \in \mathcal{G}, \|\mathbf{x}^\gamma\| \neq 0\}$. The HSPG pseudo-code (proximal-gradient variant) is given in Algorithm 3 (in the appendix A.6). We mention the enhanced variant of the standard HSPG algorithm named AdaHSPG+ Dai et al. (2023) improves upon that implementing adaptive strategies that optimize performance, focusing on better handling of complex or dynamic problem scenarios where standard HSPG may be less efficient.

## 4 Experiments

In this section, we present a comprehensive benchmark of structured sparse-inducing regularization techniques. Our evaluation covers a wide range of model architectures, datasets, regularizers, optimizers, and pruning techniques. This extensive benchmarking demonstrates that WGSEF achieves state-of-the-art performance across all these dimensions.

### 4.1 Evaluation of sparsity-inducing optimization methods

To demonstrate the performance of Algorithm 2 in terms of compression and accuracy, as compared to state-of-the-art prox-SGD-based optimization methods, we use the following well-known DNNs benchmark architectures: VGG16 Simonyan & Zisserman (2014), ResNet18 He et al. (2015), and MobileNetV1 Howard et al. (2017). These architectures were tested on the datasets CIFAR-10 Krizhevsky et al. (2009) and Fashion-MNIST Xiao et al. (2017). While WGSEF is a regularizer rather than an optimization algorithm, we benchmark it versus the Group Lasso regularizer using various optimization algorithms that are well-known for their superior performance with group sparsity inducing regularization. All experiments were conducted over 300 epochs. For the first 150 epochs, we employed Algorithm 2, and for the leftover epochs, we used the HSPG with the WGSEF acting as a regularizer (i.e., Algorithm 3). Experiments were conducted using a mini-batch size of $b = 128$ on an A100 GPU. The coefficient for the WGSE regularizer was set to $\lambda = 10^{-2}$. In Table 1, we compare our results with those reported in Dai et al. (2023). The primary metrics of interest are the neural network group sparsity ratio, and the prediction accuracy (Top-1). Notably, the WGSE achieves a markedly higher group sparsity compared to all other methods except AdaHSPG+, for which we obtained slightly better results. It should be mentioned that all techniques achieved comparable generalization error rates on the validation datasets.

Table 1: Comparison of state-of-the-art techniques to our WGSEF regularization technique, in terms of group-sparsity-ratio/validation-accuracy (Top-1), both in percentage, for various models and datasets. Our method provides the highest sparsity level with a comparable accuracy.

| Model | Dataset | Prox-SG | Prox-SVRG | HSPG | AdaHSPG+ | **WGSEF** |
|---|---|---|---|---|---|---|
| VGG16 | CIFAR-10 | 54.0 / 90.6 | 14.7 / 89.4 | 74.6 / 91.1 | 76.1 / 91.0 | **76.8 / 91.5** |
| | F-MNIST | 19.1 / 93.0 | 0.5 / 92.7 | 39.7 / **93.0** | 51.2 / 92.9 | **51.9** / 92.8 |
| ResNet18 | CIFAR-10 | 26.5 / 94.1 | 2.8 / 94.2 | 41.6 / 94.4 | 42.1 / **94.5** | **42.6 / 94.5** |
| | F-MNIST | 0.0 / 94.8 | 0.0 / 94.6 | 10.4 / **94.9** | 43.9 / **94.9** | **44.2 / 94.9** |
| MobileNetV1 | CIFAR-10 | 58.1 / 91.7 | 29.2 / 90.7 | 65.4 / **92.0** | 71.5 / 91.8 | **71.8** / 91.9 |
| | F-MNIST | 62.6 / 94.2 | 42.0 / 94.2 | 74.3 / 94.5 | 78.9 / **94.6** | **79.1** / 94.5 |

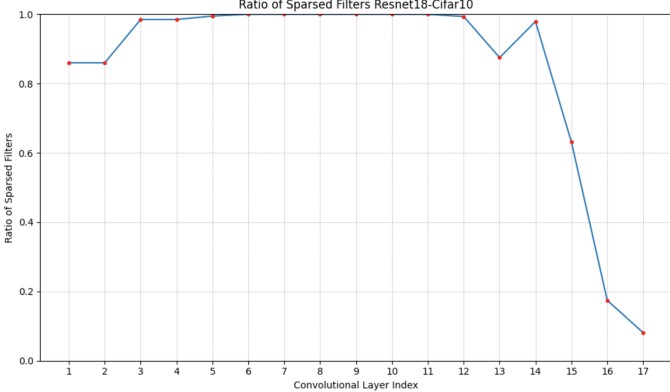

Figure 1: Ratio of the sparse filters in the convolutional layers, according to the order of the layers in the resnet18 model, as obtained by Algorithm 2, and corresponds to the experiment in row 3 of table 4.

## 4.2 Evaluation of different sparsity-inducing regularizers

In this experiment, we trained deep residual networks ResNet40 He et al. (2015) on CIFAR-10, while applying Algorithm 2, with a predefined sparsity level of 55% for all filters in all convolutional layers of the network, over 5 runs. Again, to have a fair comparison, the baseline model was trained using SGD, both with an initial learning rate of $\alpha_0 = 0.01$, regularization magnitude $\lambda = 0.03$, a batch size of 128, and a cosine annealing learning rate scheduler. Our results in Table 2 demonstrate our method's superiority over state-of-the-art structured sparsity-inducing regularizers. These results are similar to those in Bui et al. (2021), and are obtained through a grid search process that varied the magnitude of regularization w.r.t. the sparsity level. Note that the model trained using our method achieved $2.14\times$ in speedup and 47.3% reduction in FLOPs.

Table 2: Comparison of state-of-the-art structured-sparse inducing regularization methods to our WGSEF technique, in terms of filter sparsity compression and accuracy in percentage, for Resnet40 and CIFAR-10. The WGSEF regularization provides the highest sparsity level with even better accuracy.

| Method | Error | Sparsed Filters |
|---|---|---|
| Baseline (SGD) | 6.854% | 0% |
| $SGL_1$ Scardapane et al. (2017) | 7.760% | 50.7% |
| $SGL_0$ Bui et al. (2021) | 8.146% | 53.4% |
| SGSCAD Lv & Fan (2009) | 8.026% | 52.2% |
| $SGTL_1$ Bui et al. (2021) | 8.096% | 53.7% |
| $SGTL_1L_2$ Tran & Webster (2019) | 7.968% | 53.7% |
| **WGSEF** | 7.264% | 54.3% |

## 4.3 Evaluation of different state-of-the-art pruning techniques on ImageNet

In this subsection, we compare our method to the state-of-the-art pruning techniques, which are often used as an alternative for model compression during (or, post) training. We train Resent50 with ImageNet dataset, using $\lambda = 0.05$, with an initial learning rate of $\alpha_0 = 0.01$, sparsity level $k = 0.34$, and use a cosine annealing learning rate scheduler. In table 3, we compare our results to those obtained by the methods in Chen et al. (2021). We emphasize that, as mentioned later, some of the methods require several stages of training, fine-tuning, etc. Our method trains the model from scratch once, as well as OTO, and thus, this is the most fair comparison. Additionally, it should be noted that all techniques achieved comparable generalization error on the validation datasets, while our method achieved better compression performance as compared to all the other techniques.

Table 3: Comparison different pruning methods with ResNet50 for ImageNet. Our method provides the highest sparsity level, lowest total number of parameters, with a comparable accuracy in both Top-1 and Top-5 metrics.

| Method | FLOPs | Number of Params | Top-1 Acc. | Top-5 Acc. |
|---|---|---|---|---|
| Baseline | 100% | 100% | 76.1% | 92.9% |
| DDS-26 Huang et al. (2018b) | 57.0 % | 61.2 % | 71.8 % | 91.9 % |
| CP He et al. (2017) | 66.7 % | - | 72.3 % | 90.8 % |
| RRBP Zhou et al. (2019) | 45.4 % | - | 73.0 % | 91.0 % |
| SFP He et al. (2018) | 41.8 % | - | 74.6 % | 92.1 % |
| Hinge Li et al. (2020a) | 46.6 % | - | 74.7 % | - |
| GBN-60You et al. (2019) | 59.5 % | 68.2 % | 76.2 % | 92.8 % |
| ResRep Ding et al. (2021) | 45.5 % | - | 76.2 % | 92.9 % |
| DDS-26 Hu et al. (2016) | 57.0% | 61.2% | 71.8% | 91.9% |
| ThiNet-50 Luo et al. (2017) | 44.2% | 48.3% | 71.0% | 90.0% |
| RBP Zhou et al. (2019) | 43.5% | 48.0% | 71.1% | 90.0% |
| GHS Yang et al. (2019) | 52.9% | - | **76.4%** | **93.1%** |
| SCP Kang & Han (2020) | 45.7% | - | 74.2% | 92.0% |
| OTO Chen et al. (2021) | 34.5% | 35.5% | 74.7% | 92.1% |
| **WGSEF** | **34.2%** | **35.1%** | 74.2% | 92.0% |

## 4.4 Evaluation of different group structures

In this experiment, we demonstrate WGSEF's performance across various architectures, datasets, and group structures. We specifically compare WGSEF to standard training using SGD without regularization. Our aim is to demonstrate WGSEF's flexibility by showing its ability to accommodate different group structures, allowing for consideration of the input data, model architecture, hardware properties, etc. We examine the effectiveness of the WGSEF in the LeNet-5 convolutional neural network LeCun et al. (1998) (the architecture is Pytorch and not Caffe and is given in Appendix A.5), on the MNIST dataset LeCun & Cortes (2010). The networks were trained without any data augmentation. We apply the WGSEF regularization on filters in convolutional layers using a predefined value for the sparsity level $k$. Table 4 summarizes the number of remaining filters at convergence, FLOPs, and the speedups. We evaluate these metrics both for a LeNet-5 baseline (i.e., without sparsity learning), and our WGSEF sparsification technique. To ensure a fair and accurate comparison, the baseline model was trained using SGD. It can be seen that WGSEF reduces the number of filters in the convolution layers by a factor of half, as dictated by $k = 8$, while the accuracy level did not decrease. Furthermore, since the sparsification is structural, there is a significant improvement in FLOPs, as well as in the latency time of inference. Repeating the same experiment, but now constraining the number of non-pruned filters in the second convolutional layer to be at most 4 (i.e., at most quarter of the baseline), the accuracy slightly deteriorates; however, significant improvements can be observed in both the FLOPs number and the speed up, as expected. The networks were trained with a learning rate of 0.001, regularization magnitude $\lambda = 10^{-5}$, and a batch size of 32 for 150 epochs across 5 runs.

Table 4: Results of running Algorithm 2, onto redundant filters in LeNet (in the order of conv1-conv2).

| LeNet-5 (MNIST) | Error | Filter (sparsity-level) | FLOPs | Speedup |
|---|---|---|---|---|
| Baseline (SGD) | 0.84 % | 6-16 | 100 %-100 % | 1.00 ×-1.00 × |
| WGSEF | 0.78 % | 3-8 | 48.7 %-21.6 % | 2.06×-4.53 × |
| WGSEF | 0.89 % | 3-4 | 48.7 %-14.7 % | 2.06×-7.31 × |
| LeNet-5 (MNIST) | Error | Parameters | FLOPs | Speedup |
| Unstructured WGSEF | 0.76 % | 75(/150)-1200(/2400) | 68.7 %-59.2 % | 1×-1× |

In Table 5, we present the results when training both VGG16 and DenseNet40 Huang et al. (2018a) on CIFAR-100 Krizhevsky et al., while applying WGSEF regularization with a predefined sparsity level with half the number of channels for VGG16, and 60% of those for the DenseNet40. The baseline model was trained using SGD, both with an initial learning rate of $\alpha = 0.01$ and regularization magnitude $\lambda = 0.01$.

Table 5: Results of running Algorithm 2, onto redundant Channels on CIFAR-100, over 250 epochs.

| DCNN CIFAR-100 | Model | Error (%) | Pruned Channels | Overall Density |
|---|---|---|---|---|
| VGG16 | Baseline | 26.28 | $\sim 0\%$ | $\sim 0\%$ |
| | **WGSEF** | 26.46 | 50% | 41.3% |
| DenseNet40 | Baseline | 25.36 | $\sim 0\%$ | $\sim 0\%$ |
| | **WGSEF** | 25.6 | 60% | 42.8% |

## 4.5 Impact of sparsification levels on model error and training dynamics

Here, we examine the effectiveness of WGSEF in training the LeNet-5, on the **Fasion**MNIST dataset. The networks were trained without any data augmentation. We apply the WGSEF regularization on filters in convolutional layers using a predefined value for the sparsity level $k$. Table 6 summarizes the number of remaining filters at convergence, FLOPs, and the speedups. We evaluate these metrics both for a LeNet baseline (i.e., without sparsity learning), and our WGSEF sparsification technique. To be accurate and fair in comparison, the baseline model was trained using SGD. We use a learning rate equal to $1e - 4$, with a batch size of 32, a momentum 0.95, and 15 epochs.

Table 6: Results of training on **Fasion**MNIST while applying WGSEF sparsification (with $\lambda = 0.05$), onto redundant filters in LeNet-5, (in the order of conv1-conv2), and neurons in Linear layers. The baseline model was trained using SGD.

| LeNet-5 (F-MNIST) | Error | Filter (non-sparse) | FC-layers sparsity | Speedup |
|---|---|---|---|---|
| Baseline | 11.1 % | 6-16 | 9 % | $1.0 \times$-$1.0\times$ |
| WGSEF | 11 % | 3-8 | 5% | $2\times$-$4.5\times$ |
| WGSEF | 14 % | 4-6 | 62% | $1.7\times$-$6.1\times$ |
| WGSEF | 12.3 % | 2-3 | 1% | $2\times$-$7.12\times$ |

The second row of Table 6 shows that our method exhibits a significant decrease in the number of non-zero filters, specifically, half of the filters (groups of parameters) were nullified and at the same time the model performance is improved. The rest of the experiments show that there was a higher sparsification in the number of non-zero filters (groups of parameters), with only a negligible degradation in the model's accuracy. Finally, in Figure 2, we illustrate the sparsity level as a function of the epoch number, during the training of the model which corresponds to the last row of Table 6. The desired (predefined) sparsity level was rapidly attained within the first three epochs, while the model continued to improve its accuracy throughout the remaining epochs without compromising the achieved sparsity.

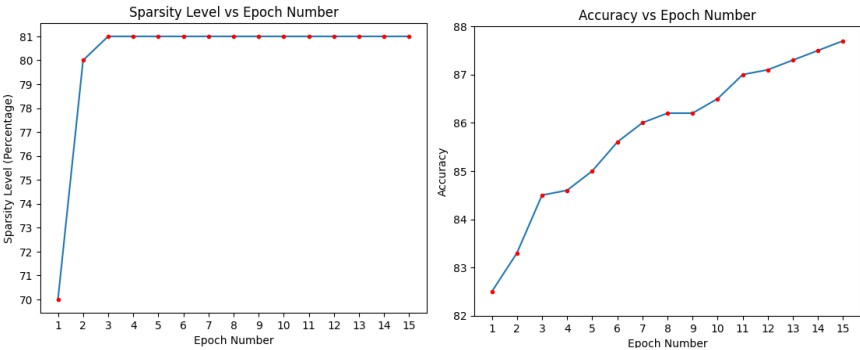

Figure 2: The graph on the left shows the level of sparseness as a function of epoch number, while the graph on the right shows the model's accuracy as a function of epoch number.

## 5 Discussion

In this study, we introduce a novel method for structured sparsification in neural network training, aiming to accelerate neural network inference and compress the neural network memory size, while minimizing the accuracy degradation (or improving accuracy). Our method utilizes a new novel regularizer, termed weighted group sparse envelope function (WGSEF), which is adaptable for pruning different specified neuron groups (e.g convolutional filter, channels), according to the unique requirements of NPU's tensor arithmetic. Mathematically, the Weighted Group Sparse Envelope Function (WGSEF) represents the optimal convex underestimator of the combined sum of weighted $\ell_2$ norms and $\ell_0$ norms. In this context, the $\ell_0$ norm is applied to the squared norms of each group. During the neural network training, the WGSE regularizer selects the $k$ most essential predefined neuron groups, where $k$ that controls the compression of the network is configurable to the trainer. Consequently, the trained neural network benefits from reduced inference latency, a more compact size, and decreased power consumption. Additionally, we show that the computational complexity of the prox operator of WGSEF, a key component in the training phase, is linear relative to the number of group parameters. This ensures that it is highly efficient and which does not constitute a bottleneck in the calculation complexity of the training process.

The experimental results show the effectiveness of WGSEF in achieving high compression ratios (reduced memory demand), and speed up in inference, with negligible compromising in accuracy. Compared to the

previous approaches, the proposed method stood out in its compression capabilities while maintaining similar network performance.

Along with the method's ability to predetermine the extent of network compression to be obtained at the training convergence, it is essential to have a prior understanding of the maximum compression level that can be applied without compromising the network's performance, and accordingly set the $k$ parameter. Naturally, it is also necessary to define the groups to which the pruning will be encouraged.

Future research could extend this work by delving into more intricate group definitions such as what could be defined in large language models. Moreover, we suggest studying a different mechanism for assessing the importance of each group being regulraized, as an alternative to the current approach which is based on the group's squared norm magnitude.

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

# A Appendix

## A.1 Proofs of results in Subsection 2.2

### A.1.1 Proof of Lemma 1

*Proof.* Let us define the axillary diagonal positive definite matrix $D^{n \times n}$, where the $D_{i,i}$ entry holds $D_{i,i} = \sqrt{d_i}, \forall i \in s_j, d_i = d_j$. Now, consider the following chain of equalities:

$$
gs_k^\star(\tilde{\theta}) = \max_{\theta \in \mathbb{R}^n} \{ \langle \tilde{\theta}, \theta \rangle - gs_k(\theta) \}
$$

$$
= \max_{\theta \in \mathbb{R}^n} \left\{ \tilde{\theta}^\top \theta - \frac{1}{2} \sum_{j=1}^m d_j \| \theta s_j \|_2^2 - \delta_{C_k}(\theta) \right\}
$$

$$
= \max_{\substack{\theta \in C_k \\ \forall i \in s_j, d_i = d_j}} \left\{ \tilde{\theta}^\top \theta - \sum_{j=1}^m d_j \cdot \| \theta s_j \|_2^2 \right\}
$$

$$
= \max_{\substack{\theta \in C_k \\ \forall i \in s_j, d_i = d_j}} \left\{ \tilde{\theta}^\top \theta - \frac{1}{2} \theta^\top D^\top D \theta \right\}
$$

$$
\overset{(a)}{=} \max_{\substack{t \in \tilde{C}_k \\ \forall i \in s_j, d_i = d_j}} \left\{ \tilde{\theta}^\top D^{-1} t - \frac{1}{2} t^\top t \right\}
$$

$$
= \max_{\substack{t \in \tilde{C}_k \\ \forall i \in s_j, d_i = d_j}} \left\{ -\frac{1}{2} \| t - D^{-1} \tilde{\theta} \|_2^2 + \frac{1}{2} \| D^{-1} \tilde{\theta} \|_2^2 \right\}
$$

$$
= \frac{1}{2} \| D^{-1} \tilde{\theta} \|_2^2 + \max_{\substack{t \in \tilde{C}_k \\ \forall i \in s_j, d_i = d_j}} \left\{ -\frac{1}{2} \| t - D^{-1} \tilde{\theta} \|_2^2 \right\}
$$

$$
= \frac{1}{2} \| D^{-1} \tilde{\theta} \|_2^2 + \max_{\substack{t \in \tilde{C}_k \\ \forall i \in s_j, d_i = d_j}} \left\{ -\frac{1}{2} \sum_{i=1}^n (t_i - (D^{-1})_{ii} \tilde{\theta}_i)^2 \right\}
$$

$$
\overset{(b)}{=} \frac{1}{2} \| D^{-1} \tilde{\theta} \|_2^2 + \max_{\substack{t \in \tilde{C}_k \\ \forall i \in s_j, d_i = d_j}} \left\{ -\frac{1}{2} \sum_{j=1}^m \| M_{s_j} (t - D^{-1} \tilde{\theta}) \|_2^2 \right\}
$$

$$
= \frac{1}{2} \| D^{-1} \tilde{\theta} \|_2^2 - \frac{1}{2} \sum_{j=m-k}^m \| M_{\langle s_j \rangle} (D^{-1} \tilde{\theta}) \|_2^2
$$

$$
\overset{(c)}{=} \frac{1}{2} \sum_{j=1}^k \| M_{\langle s_j \rangle} (D^{-1} \tilde{\theta}) \|_2^2
$$

$$
= \frac{1}{2} \sum_{j=1}^k \frac{1}{d_j} \| M_{\langle s_j \rangle} (\tilde{\theta}) \|_2^2
$$

where, $(a)$ the set $\tilde{C}_k$ is given by

$$\tilde{C}_k = \left\{\theta : \left\| \left\| (D\theta)\, s_1 \right\|^2, \left\| (D\theta)\, s_2 \right\|^2 \ldots, \left\| (D\theta)\, s_m \right\|^2 \right\|_0 \leq k\right\},$$

$(b)$ follows by the fact that the sum of the squares of the coordinates of the input vector in the support of the disjoint subset $s_1, s_2, \ldots, s_m$ that completes the index space $\bigcup_{i=1}^m s_i = [n]$ is equal to the sum of squares of the coordinates of the original vector, and $(c)$ follows by the fact that $\sum_{j=m-k}^m \left\| M_{\langle s_j\rangle}(D^{-1}\tilde{\theta}) \right\|_2^2$ is the sum of square $\ell_2$-norm of the $m - k$ disjoint subsets of $D^{-1}\tilde{\theta}$ with the smallest $\ell_2$-norm, while is the sum of squared $\ell_2$-norm of all disjoint subsets of $D^{-1}\tilde{\theta}$. $\qquad\square$

### A.1.2 Proof of Lemma 2

*Proof.* We first note that

$$\sum_{i=1}^k \left\| M_{\langle s_i\rangle}(D^{-1}\tilde{\theta}) \right\|_2 = \max_{u \in B_k} \sum_{i=1}^m u_i \left\| M_{s_i}(D^{-1}\tilde{\theta}) \right\|_2^2, \tag{13}$$

where

$$B_k \triangleq \left\{ u \in \mathbb{R}^m \mid 0 \leqslant u \leqslant e,\, e^\top u \leq k \right\}. \tag{14}$$

Now, Consider the following chain of inequalities:

$$\mathcal{GS}_k(\theta) = \max_{\tilde{\theta} \in \mathbb{R}^n} \left\{ \langle \theta, \tilde{\theta}\rangle - gs^\star(\tilde{\theta}) \right\}$$

$$= \max_{\tilde{\theta} \in \mathbb{R}^n} \left\{ \theta^\top \tilde{\theta} - \frac{1}{2} \sum_{i=1}^k \left\| M_{\langle s_i\rangle}(D^{-1}\tilde{\theta}) \right\|_2^2 \right\}$$

$$= \max_{\substack{\tilde{\theta} \in \mathbb{R}^n \\ t = D^{-1}\tilde{\theta}}} \left\{ \theta^\top Dt - \frac{1}{2} \sum_{i=1}^k \left\| M_{\langle s_i\rangle}(t) \right\|_2^2 \right\}$$

$$= \max_{t \in \mathbb{R}^n} \left\{ \theta^\top Dt - \frac{1}{2} \max_{u \in D_k} \sum_{i=1}^m u_i \left\| M_{s_i}(t) \right\|_2^2 \right\}$$

$$= \max_{t \in \mathbb{R}^n} \left\{ \theta^\top Dt - \frac{1}{2} \max_{u \in D_k} \left\{ \sum_{j=1}^m u_j \left( t^\top A_{s_j}^\top A_{s_j} t \right) \right\} \right\}$$

$$\overset{(a)}{=} \max_{t \in \mathbb{R}^n} \left\{ \theta^\top Dt + \frac{1}{2} \min_{\mathbf{u} \in D_k} \left\{ \sum_{j=1}^m (-u_j) \left( t^\top A_{s_j} t \right) \right\} \right\}$$

$$= \frac{1}{2} \max_{t \in \mathbb{R}^n} \left\{ \min_{\mathbf{u} \in D_k} \left\{ 2\theta^\top Dt - \sum_{j=1}^m u_j t^\top A_{s_j} t \right\} \right\}$$

$$\overset{(b)}{=} \frac{1}{2} \min_{\mathbf{u} \in D_k} \left\{ \max_{t \in \mathbb{R}^n} \left\{ 2\theta^\top Dt - \sum_{j=1}^m u_j t^\top A_{s_j} t \right\} \right\}$$

$$= \frac{1}{2} \min_{\mathbf{u} \in D_k} \left\{ \sum_{j=1}^m \max_{t \in \mathbb{R}^n} 2\theta^\top A_{s_j} Dt - u_j t A_{S_j} t \right\}$$

$$= \frac{1}{2} \min_{\mathbf{u} \in D_k} \left\{ \sum_{j=1}^m \max_{t \in \mathbb{R}^n} 2\sqrt{d_j}\, \theta^\top A_{s_j} t - u_j t A_{S_j} t \right\}$$

$$= \frac{1}{2} \min_{\mathbf{u} \in D_k} \left\{ \sum_{j=1}^m \phi \left( \sqrt{d_j} A_{s_j} \theta, u_j \right) \right\} \tag{15}$$

$$= \frac{1}{2} \min_{\mathbf{u} \in D_k} \left\{ \sum_{j=1}^m d_j \phi \left( A_{s_j} \theta, u_j \right) \right\},$$

where $(a)$ follows by the fact that $A_{s_j}$ is a self-adjoint matrix, and $(b)$ follows from the fact that the objective function is concave w.r.t. $\tilde{y}$ and convex w.r.t $u$, and the MinMax Theorem v. Neumann (1928). □

### A.1.3 Proof of Corollary 2.1

*Proof.* Directly by expression (34). □

### A.2 Proofs of results in Subsection 2.3

### A.2.1 Proof of Lemma 3

*Proof.* Recall that

$$v = \text{prox}_{\lambda \mathcal{GS}_k}(t) = \operatorname*{argmin}_{\theta \in \mathbb{R}^n} \left\{ \lambda \mathcal{GS}_k(\theta) + \frac{1}{2} \|\theta - t\|_2^2 \right\}.$$

Using Lemma 2, the above minimization problem can be written as

$$\min_{\mathbf{u} \in D_k} \min_{\theta \in \mathbb{R}^n} \left\{ \Phi(\theta, u, t) \equiv \frac{\lambda}{2} \sum_{j=1}^m d_j \phi \left( A_{s_j} \theta, u_j \right) + \frac{1}{2} \|\theta - t\|_2^2 \right\}$$

$$= \min_{\mathbf{u} \in D_k} \min_{\theta \in \mathbb{R}^n} \left\{ \Phi(\theta, u, t) \equiv \frac{\lambda}{2} \sum_{j=1}^m d_j \phi \left( A_{s_j} \theta, u_j \right) + \frac{1}{2} M_{s_j} (\theta - t)_2^2 \right\}$$

$$= \min_{\mathbf{u} \in D_k} \min_{\theta \in \mathbb{R}^n} \left\{ \Phi(\theta, u, t) \equiv \frac{\lambda}{2} \sum_{j=1}^m d_j \phi \left( A_{s_j} \theta, u_j \right) + \frac{1}{2} (\theta - t)^\top A_{s_j} (\theta - t) \right\}. \tag{16}$$

Solving for $\theta$, we get that for any $j \in [m]$, if $d_j A_{s_j} \theta \geq 0$ then,

$$\frac{d_j \lambda A_{s_j} \hat{\theta}}{u_j} + A_{s_j} (\hat{\theta} - t) = 0$$

$$A_{s_j} \hat{\theta} \left( \frac{d_j \lambda}{u_j} + 1 \right) - A_{s_j} t = 0$$

$$A_{S_j} \left( \hat{\theta} \left( \frac{d_j \lambda}{u_j} + 1 \right) - t \right) = 0,$$

meaning that,

$$v_i = \frac{t_i u_j}{\lambda d_j + u_j}, \ j \in [m], i \in s_j, \tag{17}$$

or, equivalently,

$$A_{s_j} v = \frac{u_j A_{s_j} t}{\lambda d_j + u_j}, \ j \in [m]. \tag{18}$$

Next, we show that $u$ is the minimizer of the problem $\min_{\mathbf{u} \in D_k} \Phi(\theta, u, t)$. Equation equation 18 also holds when $u_j = 0$, since in that case, $v_i = \hat{\theta}_i = 0$, for all $i \in s_j$. Plugging equation 18 in $\Phi$, yields,

$$\Phi(\hat{\theta}, u, t) = \frac{1}{2} \sum_{j=1}^m d_j \left( \lambda \frac{\hat{\theta}^\top A_{s_j} \hat{\theta}}{u_i} \right) + \frac{1}{2} \|\hat{\theta} - \mathbf{t}\|_2^2 \tag{19}$$

$$= \frac{1}{2} \sum_{j=1}^{m} \left( \lambda d_j \frac{\hat{\theta}^\top A_{s_j} \hat{\theta}}{u_j} + \|A_{s_j} \left( \hat{\theta} - t \right)\|_2^2 \right)$$

$$= \frac{1}{2} \sum_{j=1}^{m} \left( \lambda d_j \frac{u_j^2 t^\top A_{s_j} t}{u_j \left( \lambda d_j + u_j \right)^2} + \left\| \frac{\lambda d_j A_{s_j} t}{\lambda d_j + u_j} \right\|_2^2 \right)$$

$$= \frac{1}{2} \sum_{j=1}^{m} \left( \lambda d_j \frac{u_j t^\top A_{s_j} t}{\left( \lambda d_j + u_j \right)^2} + \frac{(\lambda d_j)^2 t^\top A_{s_j} t}{\left( \lambda d_j + u_j \right)^2} \right)$$

$$= \frac{\lambda}{2} \sum_{j=1}^{m} d_j \frac{t^\top A_{s_j} t}{\lambda d_j + u_j}$$

$$= \frac{\lambda}{2} \sum_{j=1}^{m} \phi \left( \sqrt{d_j} A_{s_j} t, \lambda d_j + u_j \right), \tag{20}$$

which concludes the proof. $\qquad\square$

### A.2.2 Proof of Corollary 3.1

*Proof.* Assigning a Lagrange multiplier for the inequality constraint $\mathbf{e}^T \mathbf{u} \leq k$ in problem (16), we obtain the Lagrangian function

$$L(\mathbf{u}, \mu) = \sum_{j=1}^{m} \left( \phi \left( \sqrt{d_j} A_{s_j} t, \lambda d_j + u_j \right) + \mu u_j \right) - k\mu.$$

Therefore, the dual objective function is given by

$$q(\mu) \equiv \min_{\mathbf{u}: 0 \leq \mathbf{u} \leq \mathbf{e}} L(\mathbf{u}, \mu) = \sum_{j=1}^{m} \varphi_{b_j, \alpha_j}(\mu) - k\mu, \tag{21}$$

for, $b_j = \left\| \sqrt{d_j} A_{s_j} t \right\|_2$ and $\alpha_j = \lambda d_j (> 0)$, where for any $b \in \mathbb{R}$ and $\alpha \geq 0$, the function $\varphi_{b,\alpha}$ is defined in Beck & Refael (2022) by

$$\varphi_{b,\alpha}(\mu) \equiv \min_{0 \leq u \leq 1} \{ \phi(b, \alpha + u) + \mu u \}, \quad \mu \geq 0. \tag{22}$$

Thus, the dual of problem (9) is the maximization problem

$$\max \{ q(\mu) : \mu \geq 0 \} \tag{23}$$

A direct projection of Lemma (Beck & Refael, 2022, Lemma 2.4) is that if $\tilde{\mu} > 0$, the function $\mathbf{u} \mapsto L(\mathbf{u}, \tilde{\mu})$ has a unique minimizer over $\{ \mathbf{u} \in \mathbb{R}^m : \mathbf{0} \leq \mathbf{u} \leq \mathbf{e} \}$ given by $u_j = \varphi'_{b_j, \alpha_j}(\tilde{\mu})$, where it was shown that

$$\varphi_{b_j, \alpha_j}(\mu) = \begin{cases} \frac{b_j^2}{\alpha_j + 1} + \mu, & \sqrt{\mu} \leq \frac{|b_j|}{\alpha_j + 1} \\ 2|b_j| \sqrt{\mu} - \alpha_j \mu, & \frac{|b_j|}{\alpha_j + 1} < \sqrt{\mu} < \frac{|b_j|}{\alpha_j}, \\ \frac{b^2}{\alpha_j}, & \sqrt{\mu} \geq \frac{|b_j|}{\alpha_j}, \end{cases}$$

for $b > 0$, otherwise 0, and the minimizer is given by

$$u(\mu^*) = \begin{cases} 1, & \sqrt{\mu} \leq \frac{|b_j|}{\alpha_j + 1}, \\ \frac{|b_j|}{\sqrt{\mu}} - \alpha_j, & \frac{|b_j|}{\alpha_j + 1} < \sqrt{\mu} < \frac{|b_j|}{\alpha_j}, \\ 0, & \sqrt{\mu} \geq \frac{|b_j|}{\alpha_j}. \end{cases}$$

Problem (23), is concave differentiable and thus the minimizer $\tilde{\mu}$ holds $q'(\tilde{\mu}) = 0$, meaning

$$q'(\tilde{\mu}) = \sum_{j=1}^{m} u_j(\mu) - k = 0.$$

We observe that for any $j \in [m]$ the functions $u_j(\mu)$ are monotonically continuous nonincresing, and therefore utilizing Lemma (Beck & Refael, 2022, Lemma 3.1) $\mu^* = \frac{1}{\eta^2}$ is the a root of the nondecreasing function,

$$g_t(\eta) \equiv \sum_{j=1}^{m} u_j(\eta) - k.$$

Note that for

$$g_t \left( \frac{\lambda \cdot min_{j \in [m]}\{d_j\}}{\left\| \sqrt{d_1} \|A_{s_1} t\|_2, \sqrt{d_2} \|A_{s_2} t\|_2, \ldots, \sqrt{d_m} \|A_{s_m} t\|_2 \right\|_\infty} \right)$$
$$= \sum_{i=1}^{m} 0 - k < 0,$$

while

$$g_t \left( \frac{\lambda \|d_1, d_2, \ldots, d_m\|_\infty + 1}{\left\| M_{\langle s_m \rangle} \left( \sqrt{d_j} t \right) \right\|_2} \right) = \sum_{i=1}^{m} 1 - k > 0.$$

Now, applying Lemma (Beck & Refael, 2022, Lemma 3.2), we deduce that $u_j(\mu)$, can be divided into the sum of the two following functions,

$$v_j(\eta) \equiv |\eta|b_j| - \alpha_j|, w_j(\eta) \equiv 1 - |\eta|b_j| - (\alpha_j + 1)|, j \in [m],$$

and thus $g_t$ can be reformulated as follows

$$g_{\mathbf{t}}(\eta) = \frac{1}{2} \sum_{j=1}^{m} v_j(\eta) + \frac{1}{2} \sum_{j=1}^{m} w_j(\eta) - k.$$

$\square$

### A.3 Proof of Corollary 3.2

*Proof.* The proof follows from (Yun et al., 2021, Corollary 1), by taking $C_t = \mathbf{0}$ and $\delta = 1$. $\square$

### A.4 Discussion on the selection of the optimization method

The general training problem we are solving is of the form

$$\min_{\mathbf{x} \in \mathbb{R}^n} f(\mathbf{x}) + h(\mathbf{x}),$$

where $f = \frac{1}{N} \sum_{i=1}^{N} f_i : X \to \mathbb{R}$ is continuously differentiable, but possibly nonconvex, and $h$ is a convex function, but possibly nonsmooth. For practical reasons, we cannot store the full gradient $\nabla f(\mathbf{x})$. Hence, we would like to use a stochastic gradient type algorithm. However, such a structure posses several difficulties from an optimization perspective, as most research of stochastic first-order algorithms does not account for both nonconvex smooth term and a nonsmooth convex regularize. In Ghadimi et al. (2016) the authors provide an analysis of a simple stochastic proximal gradient algorithm, where at each iteration a minibatch of weights is updated using a gradient step followed by a proximal step. This algorithm is proved to converge, however, the rate of convergence depends heavily on the minibatch size, and, in fact, for reasonably sized minibatches it will not converge. J Reddi et al. (2016) proposes variance-reduction type algorithms, but since these extend SAGA Defazio et al. (2014) and SVRG Johnson & Zhang (2013) to the nonconvex and nonsmooth setting, they require storing the gradient for each sample (SAGA) which requires $\mathcal{O}(Nn)$ storage, or recomputing the full gradient every $s \geq N$ iterations (SVRG), which is undesirable for training neural networks.

The ProxSGD algorithm appears appealing to our problem as it allows for momentum. While the algorithm has a convergence guarantee, the authors do not provide the rate, making it less appealing, given the known

issue with the minibatch size. We have found the most suitable optimization algorithm to be ProxGen Yun et al. (2021), as it can accommodate momentum, and also has a proven convergence rate with a fixed and reasonable minibatch size of order $\Theta(\sqrt{N})$. Next, we provide a convergence guarantee for Algorithm 1, as given in Yang et al. (2020a). The convergence is in terms of the subdifferential defined as follows.

**Definition 2** (Fréchet Subdifferential)**.** *Let $\varphi$ be a real-valued function. The Fréchet subdifferential of $\varphi$ at $\bar{\theta}$ with $|\varphi(\bar{\theta})| < \infty$ is defined by*

$$\widehat{\partial}\varphi(\bar{x}) \triangleq \left\{ \theta^* \in \Omega \;\middle|\; \liminf_{\theta \to \bar{\theta}} \frac{\varphi(\theta) - \varphi(\bar{\theta}) - \langle \theta^*, \theta - \bar{\theta} \rangle}{\|\theta - \bar{\theta}\|} \geq 0 \right\}.$$

### A.4.1 Assumptions

The following assumptions in terms of the objective function $f$ and the algorithm parameters are required:

**(C-1)** ($L$-smoothness) The loss function $f$ is differentiable, $L$-smooth, and lower-bounded:

$$\|\nabla f(x) - \nabla f(y)\| \leq L\|x - y\| \quad \text{and} \quad f(x^*) > -\infty.$$

**(C-2)** (Bounded variance) The stochastic gradient $g_t = \nabla f(\theta_t; \xi)$ is unbiased with bounded variance:

$$\mathbb{E}_\xi\big[\nabla f(\theta_t; \xi)\big] = \nabla f(\theta_t), \quad \mathbb{E}_\xi\big[\|g_t - \nabla f(\theta_t)\|^2\big] \leq \sigma^2.$$

**(C-3)** (i) Final step-vector is finite, (ii) the stochastic gradient is bounded, and (iii) the momentum parameter is exponentially decaying, namely,

$$\text{(i)} \;\; \|\theta_{t+1} - \theta_t\| \leq D, \quad \text{(ii)} \;\; \|g_t\| \leq G, \quad \text{(iii)} \;\; \rho_t = \rho_0 \mu^{t-1},$$

with $D, G > 0$ and $\rho_0, \mu \in [0, 1)$.

Based on these assumptions we can state the following general convergence guarantee.

### A.5 LeNet convolutional neural network architecture

| Layer | # filters I neurons | Filter size | Stride | Size of feature map | Activation function |
|---|---|---|---|---|---|
| Input | — | — | — | $32 \times 32 \times 1$ | |
| Conv 1 | 6 | $5 \times 5$ | 1 | $28 \times 28 \times 6$ | Relu |
| MaxPool2d | | $2 \times 2$ | 2 | $14 \times 14 \times 6$ | |
| Conv 2 | 16 | $5 \times 5$ | 1 | $10 \times 10 \times 16$ | Relu |
| MaxPool2d | | $2 \times 2$ | 2 | $5 \times 5 \times 16$ | |
| Fully Connected 1 | — | — | — | 120 | Relu |
| Fully Connected 2 | — | — | — | 84 | Relu |
| Fully Connected 3 | — | — | — | 10 | Softmax |

### A.6 Additional algorithm

---

**Algorithm 3: General Stochastic Proximal Gradient Method**

---

**Input:** Stepsize $\alpha_t, \{\rho_t\}_{t=1}^{t=T} \in [0,1)$, regularization parameter $\lambda$, switch condition $\mathcal{S}$, projection threshold $\epsilon$.

**Initialization:** $\boldsymbol{\theta}_1 \in \mathbb{R}^n$ and $\mathbf{m}_0 = \mathbf{0} \in \mathbb{R}^n$.

**for** iteration $t = 1, \ldots, T$:
    **if** condition $\mathcal{S}$ is **not** satisfied:
        Apply Algorithm 2
    **else**:
        Draw a minibatch sample $\xi_t$
        $\mathbf{g}_t \longleftarrow \nabla f(\boldsymbol{\theta}_t^{\mathcal{I}^{\neq 0}}; \xi_t) + \nabla h(\boldsymbol{\theta}_t^{\mathcal{I}^{\neq 0}}; \xi_t)$
        $\tilde{\boldsymbol{\theta}}_t^{\mathcal{I}^{\neq 0}} \longleftarrow \boldsymbol{\theta}_{t-1} - \alpha_t \mathbf{g}_t, \ \tilde{\boldsymbol{\theta}}_t^{\mathcal{I}^0} \longleftarrow \mathbf{0}$
        **for** each group $\gamma \in \mathcal{I}^{\neq 0}$:
            **if** $\langle \tilde{\boldsymbol{\theta}}_t^{\gamma}, \boldsymbol{\theta}_{t-1}^{\gamma} \rangle < \epsilon \|\boldsymbol{\theta}_{t-1}^{\gamma}\|^2$:
                $\tilde{\boldsymbol{\theta}}_t^{\gamma} \longleftarrow \mathbf{0}$
        $\boldsymbol{\theta}_{t+1} \longleftarrow \tilde{\boldsymbol{\theta}}_t^{\gamma}$
**return** $\boldsymbol{\theta}$

---

