# OpenReview forum: "Learning $k$-Level Structured Sparse Neural Networks Using Group Envelope Regularization"
_TMLR — Accepted by TMLR_

### Review · Reviewer_smJw · 2024-06-05

**Summary Of Contributions:**

The paper develops a new group sparse regularization to compress neural network models without compromising its accuracy.

**Audience:**

Yes

**Broader Impact Concerns:**

No concerns on the ethical implications of the work

**Claims And Evidence:**

Yes

**Requested Changes:**

See comments in the Weakness part.

**Strengths And Weaknesses:**

Strength:
1) The paper is well-written and easy to follow, with extensive review of related works. The key ideas are articulated well.
2) The proposed method shows strong performance in numerical tests.

Weakness:
1) The paper appears to build on the previous work Beck & Refael (2022), extending the idea of sparse envelope function (SEF) to group sparse envelop function (WGSEF) to enforce group sparsity and applying to neural network compression problem. While this extension may not be a trivial task, I am not convinced that the paper introduces significant technical novelty. There seems a notable similarity between the development of convex relaxation and proximal mapping for WGSEF (Section 2) and SEF in the aforementioned paper.
2) The authors mentioned that the proposed approach aims to reduce memory demand, power consumption and bridge the DNN hardware deployment challenges. Please elaborate (and quantify if possible) how the results in the numerical tests relate to this goal?
3) It is not clear how the neurons are grouped in most of the numerical tests. Are they all according to convolutional filter? Is the performance of WGSEF sensitive to the definition of groups? An ablation study would be helpful.
4) The authors presented multiple tests in the Experiments section, but there is only one test comparing the method with the SOTA pruning techniques. Here, the performance of WGSEF and the existing method OTO were shown very close. Additional evaluations would be needed to firmly establish the advantage of WGSEF over other baselines.

---

> ### Author Response · Authors · 2024-06-27
>
> We would like to thank the reviewer for this thoughtful and thorough review! We believe that we have addressed all of his/her concerns and questions in our response below (adhering to the section numbering provided by the reviewer).
>
> 1. The regularization scheme suggested by Beck & Refael (2022), namely SEF, is tailored for convex optimization problems derived from the motivation that the sought-solution model should be sparse (prior knowledge - sparse model recovery). The limitation of the SEF regularization is that the resulting unstructured sparsity cannot efficiently leverage the tensor-level computational primitives that enable the efficiency of modern computing hardware.
> This has motivated us to propose a generalized "group version" of the SEF that enforces sparsity not at the level of individual entries, but at the level of groups of variables. This version is also tailored to tackle non-convex optimization. This means that entire hardware components (such as those implementing rational filters, etc.) can be completely nullified, leading to significant savings in calculation time, inference energy, battery usage, and memory.
> First, we needed to find a way to extend SEF to a group structure. We considered various norms to impose on the groups' squared norms, which we realized created a degenerated SEF instance. After finalizing the use of the L2 norm on the groups, we still needed to develop an optimization algorithm tailored for non-convex optimization (model training problem), as detailed in Section 3. Additionally, adding group weights was not straightforward. Initially, we used fixed-size groups, but since this was limiting, we extended it to allow for arbitrary group sizes. All these points are reflected in the proofs.
>
> 2. The percentage of nullified-parameters (in groups) corresponds precisely to the percentage of memory required, w.r.t the total amount of learnable parameters. As shown in the experiments section, the reduction in the number of non-zero parameters is substantial, leading to a dramatic decrease in memory usage, even for storing the models trained with this method.
> The inference time acceleration significantly eases hardware deployment challenges, enhancing the user experience. Without this reduction in inference time, the latency might not have been acceptable to users. The speed improvement in inference is demonstrated in Tables 4 and 6 in the revised version (4 and 5 in before the revision).
> When all parameters in hardware components (those that implement for example filters, channels etc) are completely nullified, it is known that energy transfer to those components can be avoided, thereby saving battery, especially in end devices. Since the amount of energy varies between different processors and among different hardware components within each processor, quantifying this variation in experiments is challenging (might be quantified by FLOPS subsection 4.4). However, this difficulty is consistent with claims made in previous studies.
>
> 3. In the experiments section, groups were defined as filters and channels (please refer to subsection 4.4). This choice is natural for the types of models used in our comparisons and is consistent with previous studies.
>
> 4. We conducted the same comparisons used in previous studies on state-of-the-art pruning methods. Our study affirmed the superiority of WGSE over these methods, except for OTO, which showed similar results. This similar performance between WGSE and OTO was also observed in experiments with smaller models.
>
> In the revised version, we have edited the experiments section to enhance clarity and refine the motivations driving the experiments.
>
> Thanks.

---

### Review · Reviewer_kBUo · 2024-06-12

**Summary Of Contributions:**

The reviewed work is concerned with the sparsity inducing training of neural networks.

Arguably the most common approach is $L^1$ regularization "Lasso." In some circumstances, this regularization can be improved upon by using the "elastic net regularizer" that consists of a sum of $L^1$ and $L^2$ regularization. Recently, an alternative approach was introduced by Beck and Refael that instead computes the bidual of the regularizer $\|\cdot\|_{L^2} + \|\cdot\|_{L^2}$.

A limitation of all the above regularization schemes is that the resulting unstructured sparsity can not efficiently leverage the tensor-level computational primitives that enable the efficiency of modern computing hardware. This has motivated previous authors to propose "group versions" that enforce sparsity not on the level of individual entreis, but on the level of groups of variables, thus enabling the use of efficient batched operations while benefitting from sparsification.

The reviewed work proposes a group version of the bidualization approach to be used in the sparse training of neural networks. The authors prove a theoretical result on the convergence of the resulting proximal gradient descent algorithm and show its practical performance in the sparsity inducing training of neural networks

**Audience:**

Yes

**Broader Impact Concerns:**

No concerns

**Claims And Evidence:**

Yes

**Requested Changes:**

No request for changes, although I recommend some more spell/typo checking

**Strengths And Weaknesses:**

Strengths:
1. The method is well motivated
2. The paper in fairly well written
3. Empirical results seem good

Weakness:
1. It is a relatively straightforward combination of existing methods and thus not terribly creative

---

> ### Author Response · Authors · 2024-06-27
>
> We would like to thank the reviewer for this thoughtful and thorough review! We are glad that he/she finds our paper well-motivated and written.
>
> As for the weakness, we would like to emphasize that we believe that our contribution lies in the following points:
>
> Introduces a novel combination: Our approach combines bidualization and group regularization uniquely for structured sparsity-inducing in the training of neural networks, which hasn't been explored in this context before.
>
> Practical relevance: By focusing on group sparsity, our method efficiently leverages tensor-level computational primitives, addressing the limitations of previous unstructured sparsity versions (SEF). In addition, the WGSEF has advantages over other structured sparsity methods in computational costs and performance (accuracy).
>
> Theoretical insights: We provide rigorous theoretical results on the convergence of our proposed proximal gradient descent algorithm, adding valuable understanding to the method, and showing that the non-convex training problem coverage to a (local) minima.
>
> Empirical performance: Our experiments demonstrate significant practical benefits in terms of structured sparsity inducing and computational efficiency, resulting in low memory requirements (compression), low inference latency, and (almost) negligible accuracy degradation, which are crucial for modern hardware deployment. This suggests a high potential for practical applications.
>
> As kindly suggested, we have corrected the typos and spelling in the revised version.
>
> Thanks

---

### Review · Reviewer_ah6q · 2024-06-23

**Summary Of Contributions:**

This paper introduces a novel approach for training sparse neural networks. Specifically, the approach starts with a dense network and employs a new regularization technique called the Weighted Group Sparse Envelope Function (WGSEF). This technique dynamically nullifies groups of neurons throughout the training process. The authors also develop new proximal-gradient-based optimization algorithms to handle the WGSEF regularization effectively. By the end of the training process, this method successfully achieves a structurally sparse network that meets a predefined sparsity level. Experimental results demonstrate that the proposed method outperforms traditional sparsity-inducing techniques, such as regularization or pruning, by achieving lower computational costs and higher sparsity levels while maintaining comparable accuracy.

**Audience:**

Yes

**Claims And Evidence:**

Yes

**Requested Changes:**

1. It is recommended to conduct the experiments within a unified framework. While it may be challenging to accommodate all the settings described in this paper into a single framework, any effort would be beneficial.

2. The paper should undergo thorough proofreading to correct typos and unclear specifications.

**Strengths And Weaknesses:**

## Strength

The focus on training sparse neural networks addresses a critical challenge in the field of machine learning. Modern neural networks are computationally intensive and inefficient in terms of parameter usage. The proposed method of this paper trains the model from scratch only once to yield structurally sparse networks. Compared to post-training pruning techniques and unstructured methods, this technique can substantially reduce both training and inference costs. This suggests a high potential for practical applications.

## Weakness
1. The experiments are not conducted in a unified framework. Various models were trained and evaluated under different conditions. For instance,  train ResNet40 on CIFAR-10 compared with other regularizers, train Resent50 on ImageNet compared with pruning techniques, and train LeNet-5 on MNIST/F-MNIST compared with SGD). This may affect the consistency and comparability of the results.

2. The presentation of this paper can be improved. There are various typos and unclear specifications, for examples:
- Page 2: "selects the most essential $k \leq m$ predefined groups", but $m$ is not previously defined.
- Page 4: It is not clear why $\delta_{C_k}(\Theta)$ takes the whole parameter space $\Theta$ as input.
- Page 5, Remark 1: $s_k(\theta)$ is defined to be a function depending on the scalar $k$, not related to the index set $s_j$. This creates great confusion.
- Page 10, Table 3: Why the *numbers of parameters* for half of the methods are not presented?

---

> ### Author Response · Authors · 2024-06-27
>
> We would like to thank the reviewer for this thoughtful and thorough review! We believe that we have addressed his/her concerns and questions in our response below (adhering to the section numbering provided by the reviewer).
>
> 1. Thanks to the reviewer's comments and to improve the structure of the experiments section, we made the following changes:
> (i) We added an introduction paragraph to the experiments section in the revised version. (ii) We further clarified the motivations of the experiments in each subsection. (iii) We reorganized the last two subsections. Now, there is a separate subsection for the evaluation of the model accuracy with different structured group sparsity choices (Table 4 and Table 5). Additionally, we included another subsection for assessing model accuracy at different sparsity levels and its training dynamics (Table 6 and Figure 2).
> We hope this clarifies better our evaluation framework.
>
> Logic of benchmarks
>
> In the experiment section, we aim to provide a comprehensive benchmark of the structured sparse-inducing regularization techniques. Our evaluation spans a broad range of model architectures, datasets, regularizers, optimizers, and pruning techniques. Through this extensive benchmarking, we demonstrate that WGSEF achieves SOTA or near-SOTA performance across all these dimensions.
> While WGSEF is a regularizer rather than an optimization algorithm, we benchmark it versus the Group Lasso regularizer using various optimization algorithms (such as HSPG, and HSPG+) that are well-known for their superior performance with group sparsity-inducing regularization. This approach provides a comprehensive view of SOTA methods and allows us to highlight the strengths of WGSEF compared to the most straightforward and commonly used group sparsity-inducing regularizers. This initial comparison, presented in Table 1, offers our first perspective on WGSEF's efficiency in terms of accuracy and compression ratio.
>
> Following, we compare WGSEF with other structured sparsity-inducing SOTA regularizers. The results, shown in Table 2, consistently demonstrate that WGSEF achieves superior accuracy and compression ratio. Additionally, we evaluate WGSEF's performance vs pruning techniques, which are often used as an alternative for model compression during (or post) training. Our findings, presented in Table 3, indicate that WGSEF outperforms these pruning techniques in both maintaining model accuracy and achieving significant group parameter reduction.
>
> In the subsequent analysis, we demonstrate WGSEF's performance across various architectures, datasets, and group structures. In this context, WGSEF is compared to non-regularized training using SGD. The results reveal that WGSEF maintains model accuracy comparable to the non-regularized case while nullifying up to 85% of parameter groups. This reduction leads to a sevenfold decrease in FLOPs (and inference latency). In contrast, the SEF regularizer, although producing a sparse model, does not offer a hardware-aware structure, thus failing to provide computational and compression benefits. This highlights WGSEF's unique advantage in structured sparsity. Table 4 presents the resulting structured sparsity when convolutional filters are defined as groups, while Table 5 displays structured sparsity when convolutional channels are the groups. This again shows the flexibility of WGSEF, which is able to accommodate different group structures, allowing consideration of the input data, model architecture, hardware properties, and more.
>
> Lastly, in Table 6, we illustrate how different sparsification levels impact model error levels, providing further insight into WGSEF's robustness performance. We also show how sparsity and accuracy evolve over training epochs: where the predefined level of sparsity is rapidly attained, while the model continued to improve its accuracy throughout the remaining epochs without compromising the achieved sparsity (as shown in Figure 2).
>
> 2. As kindly suggested, we thoroughly proofread the paper. We would now like to clarify the last two examples provided by the reviwer:
>
> Remark 1: $s_k(\theta)$ is defined to be a function depending on the scalar $k$, not related to the index set $s_j$.
> -This is true; however, $gs_k(\theta)$ accepts $\theta$ as a parameter and is defined by $k$, the number of groups, where the groups $s_i$ for $i \in [m]$ are predefined. Therefore, mentioning the groups intended for an understanding of $gs_k(\theta)$ (and not $s_k(\theta)$). Thus, when $gs_k()$ is clarified, it illustrates how $s_k()$ is generalized by $gs_k()$, where the number of groups is equal to the number of parameters (singleton groups).
>
> Table 3: Why the numbers of parameters for half of the methods are not presented?
> -The table, originally from the OTO paper (by Tianyi Chen 2021), compared various methods. We have added the performance of our method to this table to provide a complete comparison of all the methods as previously done.
>
> Thanks.

---

### Decision · Action_Editor_3uYh · 2024-08-05

**Recommendation:** Accept with minor revision

**Comment:**

The reviewers are in agreement that the paper can be accepted for TMLR, with the paper's main claims being sufficiently supported by both theoretical and empirical evidence, and the results clearly being of interest to some ML researchers.  At least two reviewers had some concerns regarding the novelty, but the authors gave reasonable responses regarding this, and in any case, novelty is not a main factor for TMLR.  No concerns were raised regarding correctness.

The reviewers didn't list any required changes, but the authors may want to look over the reviews and/or latest changes once more just in case.

I would also encourage the authors to consider the wording of the main claims, e.g.:
"The properties of the WGSEF allow to pre-define the desired sparsity level that would be achieved at the training convergence while maintaining negligible network accuracy degradation..."
This wording sounds strange, because it must be impossible to pre-define an arbitrary sparsity level without degradation (because if it's too low, there will be unavoidable degradations).  I suggest some re-phrasing to preclude potential concerns of this kind, and similarly double-checking the other main claims made.

As a minor style point, in the Appendix A headings (and possibly elswhere), replace lemma -> Lemma, section -> Section, etc.

**Audience:**

This paper is well within the scope of TMLR, combining aspects of deep learning and sparse recovery.  The results are clearly of interest to the community.

**Claims And Evidence:**

This paper studies the problem of learning sparse neural networks, with the over-arching goal of reducing the required resources (e.g., memory and power) by removing redundant/extraneous parameters.  A regularization technique WGSEF based on ground sparsity is introduced, generalizing earlier 'standard sparsity' considerations.  The mains claims concerns the technique's effectiveness and efficiency, and this is backed up via experiments comparing against baselines and existing methods, and theory including convergence guarantees under suitable conditions.  The assumptions of the theory are laid out clearly in (C-1)--(C-3).